# Theoretical Modeling of Redox Potentials of Biomolecules

**DOI:** 10.3390/molecules27031077

**Published:** 2022-02-05

**Authors:** Cheng Giuseppe Chen, Alessandro Nicola Nardi, Andrea Amadei, Marco D’Abramo

**Affiliations:** 1Department of Chemistry, Sapienza University of Rome, 00185 Rome, Italy; giuseppe.chen@uniroma1.it (C.G.C.); alessandronicola.nardi@uniroma1.it (A.N.N.); 2Department of Chemical and Technological Sciences, Tor Vergata University, 00133 Rome, Italy; andrea.amadei@uniroma2.it

**Keywords:** redox potentials, DNA, proteins, theoretical-computational chemistry

## Abstract

The estimation of the redox potentials of biologically relevant systems by means of theoretical-computational approaches still represents a challenge. In fact, the size of these systems typically does not allow a full quantum-mechanical treatment needed to describe electron loss/gain in such a complex environment, where the redox process takes place. Therefore, a number of different theoretical strategies have been developed so far to make the calculation of the redox free energy feasible with current computational resources. In this review, we provide a survey of such theoretical-computational approaches used in this context, highlighting their physical principles and discussing their advantages and limitations. Several examples of these approaches applied to the estimation of the redox potentials of both proteins and nucleic acids are described and critically discussed. Finally, general considerations on the most promising strategies are reported.

## 1. Introduction

The oxidation–reduction potential of biomolecules such as proteins and DNA is an important chemical property that controls numerous chemical processes. Indeed, the ability of proteins to accept or lose electrons determines the fate of chemical reactions of biological relevance of paramount importance such as photosynthesis, respiration, metabolism, and signaling processes. In proteins, the redox center is often a specific site where a transition metal—which can assume different oxidation states—can act as an electron sink or source.

For example, the large family of Cytochromes P450 shares a cysteinato-heme prosthetic group (iron (III) protoporphyrin-IX coordinated by a proximal cysteine) as the active site [1]. In Cytochromes c, one or more c-type hemes are bound to the protein domain (structurally conserved) via two sulfhydryl groups of two cysteine residues [2,3]. In the family of hydrogenases, found in many microorganisms, the redox center includes metal atoms, typically a dinuclear catalytic center (FeFe or NiFe), coordinated by CN^−^ and CO ligands, and bound to the protein through cysteic sulfur ligand(s) [4]. In bacteria and plants, blue copper proteins, which contain type I copper sites, play a significant role in the electron transport between other proteins [5]. As an example, azurin transfers electrons between cytochrome C551 and cytochrome oxidase in bacterial respiratory chains, whereas plastocyanin, which can be found in higher plants and algae, allows for the electron transport between photosystem II and I of the chloroplast [5].

In addition, many organic molecules acting as cofactors play key roles in redox reactions. For example, nicotinamide adenine dinucleotide (NAD^+^) and reduced nicotinamide adenine dinucleotide (NADH) are key cofactors for many biochemical processes such as cellular energy metabolism [6]. In fact, NAD^+^ is involved as a cofactor of the glyceraldehyde 3-phosphate dehydrogenase in the glycolysis, mediating the oxidation of the d-glyceraldehyde 3-phosphate to form d-glycerate 1,3-bisphosphate. The redox couple NADH/NAD^+^ is involved in the cellular energy metabolism, and experimental evidence have suggested that it plays a role in gene expression [7] and calcium homeostasis [8].

Other essential redox-active coenzymes are the flavin adenine dinucleotide (FAD) and the flavin mononucleotide (FMN). FAD and FMN are cofactors or prosthetic groups of the family of flavoproteins and catalyze many redox processes such as the dehydrogenation of succinate in the Krebs cycle, beta-oxidation, and degradation of amino acids. They are also important in the biosynthesis of coenzyme A, coenzyme Q, heme, and other important cofactors and hormones [9]. Electron-transfer processes involving nucleic acids are associated with gene-protection mechanisms [10] as well as the search for DNA lesions by proteins [11]. Furthermore, the possibility of regulating the redox properties of nucleic acids acting on the sequence and/or conformation suggests the use of nucleic acids as nanoscale devices [12].

Among the four DNA nucleobases, guanine (G) is the easiest to oxidize, as shown in different experimental and computational works [13,14,15,16]. The guanine base becomes easier to oxidize when embedded in G-rich sequences [17,18,19] due to the stabilization of the hole by the adjacent guanine bases. Telomeres are an important example of G-rich systems. Their characteristic sequence in vertebrates is 3${}^{\prime }$-(TTAGGG)_n_-5${}^{\prime }$ [20], and it is particularly prone to oxidation [21]. This has led to the conclusion that telomeres, found at the end of chromosomes, are able to protect genetic material from oxidation [21]. In general, the oxidation trend of the four nucleobases is well established, but, on the other hand, it is well known that the sequence affects the ionization and the reduction potentials of a single nucleobase or of several adjacent nucleobases (assuming that the hole is localized or delocalized, respectively) [19]. This is corroborated by studies on the kinetics of (positive) charge transfer, where sequence length and type were found to remarkably alter the kinetics of the processes [22,23].

Therefore, the relevance of the role of the redox potentials in bio-macromolecules as well as the experimental difficulties to provide reliable estimates of them under physiological conditions prompted numerous theoretical-computational investigations. However, due to the complexity of the environment, reliable estimates of the redox potentials of specific regions of such macro-molecules involve important issues.

In fact, although the electronic properties of small–medium-sized molecules can be accurately treated by means of quantum mechanical approaches, an accurate theoretical treatment able to couple the environmental effects felt by the redox center, including the structural–dynamical evolution of the system, with its electronic properties that still represent a challenge. In the specific case of the redox potential, the observable of interest is the free energy of the electron-transfer process, and thus a proper sampling of the system-accessible conformational space is mandatory.

To address such an issue, different theoretical treatments have been proposed in the past. The simplest one, which uses a description of the environment as a continuous dielectric, is rarely used, because the part of the system where reduction/oxidation takes place is usually embedded in an inhomogeneous environment. Therefore, in this specific context, QM/MM methods are more used, because they are able to treat the relevant part of the system at a quantum-mechanical level of detail, whereas the MM part is treated at the classical level, allowing for a proper description of the perturbation on the QM region due to the environment.

Finally, full QM treatments of the system dynamical evolution can be tentatively addressed by ab-initio molecular-dynamics methods, which explicitly treat the quantum-mechanical effects of the electrons of the systems. However, these computationally demanding approaches often suffer from a rather limited sampling in both size and time scales.

These different strategies are briefly discussed in the following section, whereas their applications to the redox potential estimates in proteins and DNA are reviewed in the last part of this work.

## 2. Theoretical-Computational Approaches

The basic equation, in the NVT ensemble, linking the standard reduction/oxidation potential ($V_{red/ox}$) with a measurable thermodynamic property is represented by the Nernst equation
(1)$$V_{red/ox}=-\frac{\Delta A_{red/ox}}{nF}-V_{ref}$$
where $\Delta A_{red/ox}$ is the (Helmholtz) free energy change upon reduction/oxidation; *F* is the Faraday constant; *n* is the number of electrons involved in the reaction; and $V_{ref}$ is the standard reference electric potential.

In the NPT ensemble, the Helmholtz free energy is replaced by the Gibbs free energy.

The computational approaches used for the estimate of the free energy of the reduction/oxidation process in proteins and nucleic acids are discussed in the following subsections (in Figure 1, a brief overview is shown).

### 2.1. Evaluation of the Free Energy by Means of Phase Space
Sampling

In classical approaches, the free-energy difference between the oxidized and the reduced states is typically computed using perturbation theory or thermodynamic integration.

In perturbation theory, the Hamiltonian of the target system is represented as a sum of a reference Hamiltonian and a perturbation term.

Considering these two states characterized by their Hamiltonians, their free-energy difference, corresponding to the standard chemical potential change ($\Delta \mu =\mu_{ox}-\mu_{red}$), can be expressed by the Zwanzig expression [24]
(2)$$\Delta A=\Delta \mu =-k_{B}T\ln{\langle e^{-\beta \Delta E}\rangle }_{red}$$
where $\Delta E=E_{ox}-E_{red}$ represents the difference of their Hamiltonian eigenvalues that is equivalent within the classical approximation to the potential energy difference between the states, $\frac{1}{\beta }=k_{B}T$ with $k_{B}$ the Boltzmann constant and *T* the absolute temperature. The subscript of the angle brackets means that averaging is performed in the reference-state ensemble (i.e., the reduced-state ensemble). Note that within the classical approximation, the difference between the two states is given by the change in their charge distribution. The accuracy of the Zwanzig equation is limited by the sampling efficiency employed to obtain the expectation value within the reference ensemble. In fact, when the two states are rather different, the sampling provided by MD simulations—usually on the order of hundreds of ns—within a given ensemble cannot typically access the phase space regions relevant for the other state, leading to inaccurate estimates of the free energy by means of the Zwanzig formula. Therefore, different strategies have been developed to overcome such a practical problem. The simplest and widely used approach is to average the Zwanzig equation over the two ensembles [25,26,27,28,29].
(3)$$\Delta \mu =-\frac{k_{B}T}{2}\ln{\langle e^{-\beta \Delta E}\rangle }_{red}+\frac{k_{B}T}{2}\ln{\langle e^{\beta \Delta E}\rangle }_{ox}=\frac{k_{B}T}{2}\ln\frac{{\langle e^{\beta \Delta E}\rangle }_{ox}}{{\langle e^{-\beta \Delta E}\rangle }_{red}}$$
often providing rather accurate estimates. A further simplification of the last equation can be obtained by using the Marcus theory [30,31,32,33], i.e., assuming, in both ensembles, exact Gaussian fluctuations of ΔE with identical variance, which is equivalent to the linear response approximation (LRA) [25,26,27,28,29,34]. In fact, by considering Gaussian distributions in both ensembles, we can write in the reduced ensemble
(4)$$\Delta \mu =-k_{B}T\ln{\langle e^{-\beta \Delta E}\rangle }_{red}\approx {\langle \Delta E\rangle }_{red}-\frac{\beta }{2}{\langle {\left(\Delta E-{\langle \Delta E\rangle }_{red}\right)}^{2}\rangle }_{red}\;$$
and in the oxidized state ensemble
(5)$$\Delta \mu =k_{B}T\ln{\langle e^{\beta \Delta E}\rangle }_{ox}\approx {\langle \Delta E\rangle }_{ox}+\frac{\beta }{2}{\langle {\left(\Delta E-{\langle \Delta E\rangle }_{ox}\right)}^{2}\rangle }_{ox}$$

Therefore, by averaging these last equations and assuming approximately identical variances, we obtain
(6)$$\Delta \mu \approx \frac{1}{2}\left({\langle \Delta E\rangle }_{red}+{\langle \Delta E\rangle }_{ox}\right)$$

The last equation provides a simple expression for the redox free energy, requiring only the evaluation in both ensembles of the average transition energy (i.e., $\Delta E=E_{ox}-E_{red}$), which can be obtained from molecular simulations. However, it is worth to note that such an approximation, relying on the severe assumption of negligible deviations from the Gaussian distribution (or at least assuming similar opposite free-energy corrections in the two ensembles) can be, in general, inaccurate.

A different strategy to overcome the practical sampling problem is the use of the thermodynamic integration method [35] based on the evaluation of the derivative of the free energy with respect to an order parameter along a path connecting the target and the reference states. Such an approach is based on the fact that a free-energy difference between two states can be expressed by means of an order parameter ρ, via
(7)$$\Delta \mu =A\left(\rho_{1}\right)-A\left(\rho_{0}\right)=\int_{\rho_{0}}^{\rho_{1}}\frac{dA}{d\rho }d\rho$$
with
$$\frac{dA}{d\rho }={\langle \frac{\partial H}{\partial \rho }\rangle }_{\rho }$$
where, again, the subscript of the angle brackets means averaging within the ensemble characterized by the ρ value. Therefore, the calculation of the free energy reduces to the estimate of the expectation value of the Hamiltonian derivative with respect to the reaction coordinate ρ at different fixed values of such a coordinate. Typically used “order-parameters” involve the charge distribution of the relevant chemical species, thus implying that the free-energy difference corresponds to a transition without polarization effects.

### 2.2. Implicit Solvent Approaches

The dielectric continuum solvation models represent a simple approach to estimate the redox-reaction free energies. In these methods, a thermodynamic cycle describes the redox free energy as a sum of different contributions: the gas-phase redox free energies and the solvation free energies of the reactants and products. The gas-phase free energy of the reaction is calculated by quantum-mechanical methods, whereas the solvation free energies of the products and reagents are evaluated considering the solvent as a polarizable dielectric.

In the effort to better describe the environmental effects on the region of the system treated at the quantum-mechanical level, the treatment of the system electrostatics may vary between different implicit solvent approaches.

Therefore, following the approach originally proposed by Tomasi [36], several groups have proposed some modifications, including, for example, the treatment of non-bulk electrostatic effects by means of a sum over empirical surface-tension terms [37], the C-PCM [38], the COSMO model [39] where the solvent is described as a conductor and the more-recent solvation model based on solute electron density (SMD) [40].

These methods have been extensively applied to relatively small molecules in solutions to compute different thermodynamic properties; however, their application is limited by the cavity definition, which in proteins and DNA is not trivial.

The redox potential of molecules by means of implicit solvent approaches is usually evaluated by (i) a thermodynamic cycle where the ionization energies are combined with the solvation energies to provide the redox potential or (ii) by using the Marcus model where the redox potential is expressed in terms of the vertical electron affinities and ionization potentials (see Figure 1).

### 2.3. Quantum-Classical (QM-CL) Approaches

Actually, the methodologies based on the coupling between extended classical descriptions of the conformational space accessible by the system (such as molecular dynamics or Monte Carlo simulations) and the treatment of a part of the system at the quantum-mechanical level are routinely used for the computation of the redox properties of biologically relevant systems.

In fact, when the portion of the system where the reduction/oxidation takes place can be determined, this kind of approach is in principle able to estimate the free energy of the process.

Nevertheless, redox-potential estimates by means of QM-CL approaches must address two main problems: the treatment of the coupling between the QM and the CL parts of the system as well as the calculation of the free energy *per se*, which typically requires an extended sampling of the conformational space due to its slow convergence. As the latter issue was already discussed above in a classical-mechanical context, we describe here the main strategies used to couple the QM and the CL regions of the system.

#### 2.3.1. QM/MM Strategies: On-the-Fly Treatment of the QM
Region

The theoretical approaches named QM/MM refer to the combination of an explicit treatment of the electronic degrees of freedom of a part of the system with a classical treatment of the remaining part of the system.

Due to the computationally demanding calculations of the electronic structure, this kind of approaches is routinely used in the context of biomolecules, where the system is typically composed by several thousands of atoms.

Here, we briefly describe the main strategies used to couple the QM with the MM part, while suggesting to the reader more exhaustive reviews on the field [41,42,43].

Although the QM part can be in principle treated at any level of theory (ranging from less-accurate semi-empirical methods to post-Hartree–Fock ab-initio approaches), the electronic properties of the QM region are usually evaluated at the density-functional-theory level. This is because DFT shows a high computational efficiency in treating large systems.

On the other hand, the MM part is usually treated using semi-empirical force fields, which are able now to provide an accurate description of the dynamical evolution even of large systems at a reduced computational cost.

The QM/MM approaches can be divided in two main schemes, called subtractive and additive.

In the subtractive scheme, the inner part is treated at both QM and CL levels, whereas the outer part is treated at the CL level only. The system energy is then estimated by summation of the inner-part QM energy and the outer-part energy subtracted by the inner-part CL energy (see, for example, the IMOMO model [44]).In this simple scheme, the coupling between the inner- and outer-parts is treated at the CL level, i.e., typically modeling the electrostatic terms as Coulomb interactions between QM- and CL-fixed atomic charges.In the additive scheme, the energy of the outer part of the system is still evaluated at CL level, but the inner part is described by using two energy terms: the QM energy of the inner part and an explicit coupling term providing the interaction between the two subparts.QM/MM methods differ by these coupling terms, which involve the description of the mutual polarization between the two subparts of the system. Briefly, such a polarization can be described as mechanical embedding, electrostatic embedding, and polarized embedding [45].These three approaches differ in how the mutual effects are treated: in the mechanical embedding scheme, the atomic charges are assigned to the QM subpart, and the interactions between the two regions are simply described by the charge–charge electrostatics. In the electrostatic embedding, the effect of the environment on the inner region is included by performing the QM calculations in the presence of the MM charges of the outer region. The polarized embedding attempts to include also the polarization of the MM charges of the outer region.The reader is referred to the extended review by Bakowies and Thiel [45], where a detailed description of these methods is reported.

Although all these treatments suffer from some shortcomings and/or computational limitations, the electrostatic embedding scheme is one of the most popular approaches as it provides a good description of the QM/MM interactions at a reasonable computational cost.

In the context of such theoretical-computational procedures, the redox potential is usually calculated by the LRA approach, briefly discussed in the previous section.

#### 2.3.2. QM/MM Strategies: *A-Posteriori* Treatment of the QM
Region

A different class of approaches is represented by *a-posteriori* schemes.

In this kind of treatment, an efficient and coherent sampling of the whole system’s configurational space is performed by a fast (classical) sampling approach, i.e., molecular dynamics. Then, the electronic properties of a specific subpart of the system are evaluated considering the effect of the environment. The simplest approach is to include in the QM calculations a part of the environment molecules or their electrostatic field to approximate the environment effects. A typical example is when the electronic properties of a small solute molecule are evaluated in solution: in this case, the QM calculation at each selected MD configuration involves the solute molecule as well as a shell of solvent molecules [46]. Although this strategy is quite computationally demanding, a sensitivity analysis on the number of frames and the size of the outer region explicitly included in the QM calculations can provide a reliable estimate of the method accuracy [47]. However, due to the high computational costs of the explicit inclusion of a large number of environment molecules in the QM calculations, such a widely used approach is typically unable to provide a complete sampling of the quantum-properties fluctuations as induced by the atomic-molecular dynamics of large solute in solution.

In this context, in 2001, a new approach, the Perturbed Matrix Method (PMM), was introduced to specifically address the issue of a proper sampling of the structural and dynamical effect of the perturbing environment on the QM region of interest [48,49,50]. Within such an approach, a molecular-dynamics simulation is coupled *a-posteriori* to QM calculations, and the perturbed energy and the electronic properties of the QM region are evaluated at each step of the MD simulation. The use of a multipolar expansion of the perturbation operator allows to obtain, at relatively low computational costs, the perturbed QM Hamiltonian at every MD frame considering the electrostatic perturbation due to all the environment molecules in the simulation box. A simple—and computationally efficient—diagonalization of the corresponding perturbed Hamiltonian Matrix leads to a set of ”perturbed” eigenvectors, which can be used to evaluate the time evolution of whatever electronic property of the QM subsystem in such a perturbed eigenstate basis set.

That is, the electronic Hamiltonian operator $\hat{H}$ of the quantum center (QC) embedded in the perturbing environment can be thus expressed *via*:(8)$$\hat{H}={\hat{H}}^{0}+\hat{V}$$
where ${\hat{H}}^{0}$ is the QC unperturbed electronic Hamiltonian (i.e., the Hamiltonian of the isolated QC), and $\hat{V}$ is the perturbation operator providing the QC–environment interaction. Within the PMM scheme, the perturbing electric field as obtained from the environment atomic charge distribution is used to obtain the perturbation operator, $\hat{V}$, *via* a multipolar expansion centered either in the QC center of mass (QC-based expansion) or at each atomic position (atom-based expansion). From the previous equation we can readily obtain the corresponding electronic Hamiltonian matrix $\tilde{H}$, i.e., its elements ${\left[\tilde{H}\right]}_{l,l^{\prime }}$, as expressed in the basis set given by the unperturbed electronic eigenstates $\Phi_{l}^{0}$ with eigenvalues $U_{l}^{0}$,
(9)$${\left[\tilde{H}\right]}_{l,l^{\prime }}=\langle \Phi_{l}^{0}|\hat{H}|\Phi_{l^{\prime }}^{0}\rangle =\delta_{l,l^{\prime }}U_{l}^{0}+\langle \Phi_{l}^{0}|\hat{V}|\Phi_{l^{\prime }}^{0}\rangle$$

By such an approach, the free energy of several molecular processes—including redox potentials—in realistic environments have been modeled so far [16,51,52,53,54,55,56].

In particular, the following expression has been used for the estimation of the redox free energies
(10)$$\mu_{red}-\mu_{ox}=\frac{k_{B}T}{2}\ln\frac{{\langle e^{-\beta \Delta U}\rangle }_{red}}{{\langle e^{\beta \Delta U}\rangle }_{ox}}$$
where ΔU is the perturbed electronic energy difference for the oxidation reaction as obtained by the PMM-MD procedure. The angular parentheses indicate the average over the MD generated reduced/oxidized (red/ox) ensemble.

### 2.4. Methods Based on a Full Quantum-Mechanical Hamiltonian

As stated before, a full QM treatment of the system and its dynamics is usually computational undoable. However, different strategies have been proposed in the past to obtain MD simulations of the atomic classical motions based on fully quantum Hamiltonians.

One of these are the Car–Parrinello molecular dynamics, where the system evolution is described, using some theoretical and computational approximations, at the QM level of details [57,58]. However, such an approach still suffers from a limited sampling and, in the specific case of redox free-energy evaluations, was often coupled to a QM/MM scheme, thus falling in the more general QM/MM approaches previously described.

A different strategy (ONIOM) based on the use of several layers [59,60] has been used to reduce the computational cost of a full QM treatment. In such an approach, the inner part of the system is treated by means of rather-accurate QM methods (e.g., DFT), whereas the outer regions are treated at lower level of details, typically using semi-empirical approaches. Although rather popular, the low accuracy of the semi-empirical methods as well as the complex description of the coupling terms between different layers lead to a difficult estimation of the accuracy of such an approach.

The quantum chemical cluster approach is an additional method developed to model chemical reactions occurring in a complex environment. The idea is to quantum-mechanically treat the active site of the enzyme (models with up to 200 atoms are often used) and the surrounding by a dielectric continuum model with the proper dielectric constant to describe polarization effects [61]. In such an approach, some degrees of freedom are locked at the periphery of the active site to prevent artificial movements that can lead to an incorrect description of the system geometry in which the process takes place [62].

### 2.5. Statistical Inaccuracies

The estimation of the method’s inaccuracies is not straightforward as several approximations—present in all of the computational procedures described above—can act as sources of errors. That is, the choice of a specific QM level of theory, the type of the functional in the DFT calculations, the size of the QM region, the choice of the cavity in implicit methods, and the length of the trajectory in QM/MM methods can affect the value of the redox potential. Therefore, it is not unexpected that only a few works report a confidence interval of the computed reported potential. However, using the available values reported in the literature [16,55,63,64,65,66,67], it is likely to assume that a plausible inaccuracy interval is between 5 and 20 kJ/mol.

## 3. Applications

In this section, we review the representative literature dealing with the calculation of the redox potential in biological contexts.

### 3.1. Redox Potential of Proteins

#### 3.1.1. Redox Potential of Copper Proteins

Copper proteins are present in numerous organisms and play a central role in many biological functions involving electron-transfer processes. In particular, blue copper proteins exchange electrons through a Cu ion site (Type I copper site) that can exist as Cu(II) and Cu(I). They are characterized by high electron-transfer rates and minimal structural changes between the two oxidation states [68,69,70].

In Type I copper proteins, a single Cu ion is coordinated by two histidine imidazoles, a cysteine thiolate and a fourth ligand (typically a methionine thioether; see Figure 2) [71]. Depending on the protein environment, they can explore a wide range of redox potential values [5,72,73]. The possibility of predicting this property is of great interest when designing new redox active proteins [74,75,76].

In the work of Li et al. [77], the issue was approached using a full quantum-mechanical treatment (using HF/6-31G*, B3LYP/6-31G*, and MP2/6-31G*). However, the high computational cost of such an approach forced the authors to introduce some significant approximations. The calculations were performed in two different conditions, one in which the solvent was described using a dielectric continuum model (IEF-PCM [78,79,80]) and one in a vacuum. In both cases, large portions of the proteins (cucumber stellacyanin, *P. aeruginosa* azurin, poplar plastocyanin, *C. cinereus* laccase, *T. ferrooxidans* rusticyanin, and human ceruloplasmin) were ignored, therefore considering only regions of ≈100 atoms near the active sites. The results were also compared to the same calculations repeated considering even smaller systems of around 50 atoms.

In the case of the larger systems, the structures were optimized using the RHF/6-31G* for Cu^+^ and ROHF/6-31G* for Cu^2+^. Since a single configuration was considered, the redox potentials estimated were approximated by the change in the electronic energies upon reduction. The authors have shown that the smaller systems do not reproduce well the experimental values, proving them as inadequate to model the Type I copper site. The calculations on the larger systems resulted in a slight improvement, although the authors point out that the observed deviations from the experimental data could be explained by the necessity to consider an even larger portion of the protein.

To properly treat the effects of both the protein environment and the solvent, Olsson et al. [81] studied the blue copper proteins plastocyanin and rusticyanin (which show different redox potentials, measured at 375 and 680 mV, respectively) using a QM/MM hybrid method. By such an approach, the region containing the active site is treated quantum-mechanically using the frozen density functional theory (FDFT) [82], where the Cu ion and its ligands are described by DFT.

An additional region of the system, which includes a few protein residues close to the copper site and water molecules (i.e., the frozen part of the system), is represented by electron densities as a way to treat the interaction between the QM and MM regions. The rest of the system is described as point charges using the ENZYMIX force-field parameters [83].

The systems, formed by the proteins and ≈500 molecules of water, were embedded in a shell of Langevin dipoles [84] and a dielectric continuum to mimic the solvation. Because of the high computational cost required, the authors concluded that it was not possible to perform the calculation on enough conformations to obtain a reliable estimate of the free energy.

To overcome such a limitation, the authors proposed the use of an empirical valence bond (EVB) reference potential [85,86]. By such an approach, the differences in the predicted reduction potentials between the two proteins ranged between 180 and 340 mV (compared to the experimental value of 305 mV) depending on how the protein was modeled, how the perturbation was treated, and which type of reference potential was considered. The same authors compared the results obtained with the values predicted using classical simulations with semi-macroscopic (PDLD/S-LRA [87]) and all-atoms approaches.

It should be noted that although the QM/MM method proposed is very elaborate, especially for a system of this complexity, the introduction of a quantum-mechanical treatment did not automatically lead to an improvement over more commonly used methods. Ultimately, this approach offers the advantages of taking into account the protein and the solvent at a molecular level, and it is able to consider multiple conformations. However, it does not completely overcome difficulties arising from the necessity of more extended sampling as well as a careful description of the system at the interface between the QM and MM regions.

Another example of the application of a QM/MM methodology to copper proteins is provided by Cascella et al. [88], where Car-Parrinello/MD computational scheme was used to study the redox properties of azurin from *Pseudomonas aeruginosa*. Due to the computationally demanding approach, the free energy of the redox process was calculated in the limit of the LRA. The Cu ion and its ligands were treated quantum-mechanically using the Perdew–Burke–Ernzerhof exchange-correlation function [89], while the rest of the system (including the rest of the protein and 8648 water molecules) was described through classical mechanics using the AMBER force field [90]. The calculated redox potential as well as the shift with respect to the redox potential of the Cu(II)/Cu(I) couple in aqueous solution was in good agreement with the corresponding experimental measurements.

In order to reduce the computationally demanding QM part of the QM/MM calculations, a recent work [63] used the QM/MM-MFEP approach [91], where the conformational space of the QM region explored during the simulation is limited to the structures along the reaction path. For each of these steps, the perturbing field generated by the MM region during the MD simulation, while considering the QM region frozen, is considered as an average in time. This strategy allowed to run relatively long simulations of ≈1 ns for each step, predicting the relative values of the redox potential of the wild-type azurin and seven mutants with good accuracy. An additional QM/MM approach, aiming to limit the computational cost still accurately describing the system dynamics, is represented by the PMM. Such an approach, employed in the study of wild-type azurin protein and two mutants (N47S/M121L and N47S/F114N/M121L) [64], was quite accurate in reproducing the shifts in redox potentials compared to the native protein. This study also highlighted the importance of an atomic model of the environment as well as of an extensive conformational sampling of the system. In fact, in the reduced ensemble, two different structures—corresponding to different values of the redox potential—contributed to the overall redox potential. Interestingly, the authors also found that only few residues are responsible for the shift in the redox potential between the WT form of the Azurin and two mutants. The analysis of the perturbation as exerted by the environment on the QC also showed that the change in the redox potential upon mutation can be ascribed to both direct effects due to the mutations inside the QC and to long-range dynamical effects affecting specific protein regions, which are able to modify the perturbation felt by the QC. An *a-posteriori* treatment of the QM region makes such an approach feasible for systems of great complexity, allowing very long simulations and, thus, an exhaustive sampling of the accessible conformational space.

The redox potential of Azurin protein was also estimated by an entirely classical approach [92]. In that work, by means of a thermodynamic integration method, the redox potentials of Azurin from *Pseudomonas aeruginosa* and some of its mutants at pH = 5 and pH = 9 were estimated. The coupling parameter ρ is associated with the charge distribution of the Cu ion and its ligands, whose variation describes the transition between the two oxidation states. When considering the average value from eight different pathways considered, the predicted value of $\Delta \mu \left(\mathrm{pH}=5\right)\;-\;\Delta \mu \left(\mathrm{pH}=9\right)$ was only 1 kJ mol^−1^ lower than the experimental value (7 kJ mol^−1^). However, as observed by the authors, the results obtained from different pathways can differ from each other (with a spread of 22 kJ mol^−1^). In particular, the authors highlight how given the same values of ρ, the values of Δμ were almost twice as large when the ρ values were considered randomly in the simulations, instead of sequentially. Considering that each step should be independent of the others, this discrepancy was attributed to the slow nature of the relaxation process of the system when the charge distribution is changed, thus requiring longer simulation times. Moreover, doubling the number of ρ values and increasing the sampling time beyond a few hundreds of picoseconds did not lead to significant improvements.

#### 3.1.2. Redox Potentials of Iron–Sulfur Proteins

Iron–sulfur clusters (see Figure 3) are common structures found in many biological systems and play a key role in electron-transfer and catalytic processes. Their redox potentials can vary depending on the protein environment, i.e., the same [4Fe-4S] cluster scans a wide range of redox potentials whether it is found in high-potential iron–sulfur proteins (HiPIPs) or in bacterial ferredoxins (Fds) [93,94,95]. Their presence in many biological systems of great interest and their high degree of tunability made them a target of several theoretical-computational studies.

In an early work [96], a broken symmetry (BS) DFT [97,98] approach, able to describe antiferromagnetic coupling, was used to calculate the redox potential of the [4Fe-4S] cluster in different proteins (e.g., HiPIPs, Fds, photosystem I, and the Fe protein of nitrogenase). The protein environment was described as point charges screened by a continuum dielectric with ϵ=4 and surrounded by a continuum dielectric with ϵ=80 representing the solvent. The active site was also screened with a dielectic of ϵ=1. A similar approach was adopted in more recent works [99,100], with the main difference of adopting an additive approach to the calculation. In that case, the Δμ associated to the reduction process of the active site (i.e., the iron–sulfur cluster and its ligands) was calculated in the gas phase using BS-DFT, while the contribution given by the change in the interaction energy with the environment between the two states was calculated using Poisson–Boltzmann continuum electrostatics and then added. This approach is valid under the condition that the environment does not affect significantly the electronic structure of the active site. By comparing the results obtained for different proteins using continuum electrostatic calculations, a good agreement with the experimental values is achieved in some cases, while in others a rather qualitative agreement is reached. The main challenge remains the dependency of the continuum solvent models on the values of the atoms’ radii and of the dielectric constant considered, which can greatly affect the results of the calculation.

Due to the complexity of these systems, only a few works attempted a QM/MM treatment of iron–sulfur proteins [101,102]. In a work by Sundararajan et al. [103], the redox potential of Rubredoxin, which presents a cluster with a single iron site, was predicted using a QM/MM approach within the ONIOM scheme. The QM region was described using DFT, while the protein and solvent environment was treated explicitly as point charges using the AMBER force field [104]. While the differences in the redox potentials between the proteins considered were correctly reproduced, the accuracy of their absolute value suffered from a rather inaccurate estimate of the entropic contribution, since the cost of the calculation did not allow to consider enough conformations of the protein.

#### 3.1.3. The NAD/FAD Redox Potential

In an earlier work, the prediction of the redox potential [105] using a QM/MM approach was attempted for FAD in medium-chain acyl-CoA dehydrogenase (MCAD) and cholesterol oxidase (CHOX; see Figure 4). In this case, the QM region, the flavin ring, was treated using the self-consistent-charge density functional tight-binding (SCC-DFTB) method [106,107], while the protein and the solvent was described using classical molecular dynamics. Such a computational strategy, using parameters obtained by means of DFT, decreases the computational cost required to study complex biochemical systems. The free energy of the electron and proton additions were calculated using a thermodynamic integration-like approach. A similar approach was previously adopted by Li et al. for the same system [108].

When comparing the redox potential relative to the full redox reaction (which includes the transfer of two electrons and two H^+^) to the values reported in the literature, a reasonable agreement was found. However, the redox potential of the semi-reaction only qualitatively reproduced the corresponding experimental data.

Nevertheless, the computational approach used by Bhattacharyya et al. [105] was able to reproduce the increased bending of the unreduced flavin ring in MCAD compared to CHOX and in aqueous solution. This structural effect, caused by the protein environment that is properly described by the MD simulations, correctly contributes to lower the reduction potential, in agreement with previous works [109].

Considering the importance of the polarization of the environment in biomolecules, highlighted in a previous work [110], Tazhigulov et al. [65] adopted a QM/BioEFP [111] approach to predict the reduction potential of cryptochrome 1 protein from *Arabidopsis thaliana*, a protein containing FAD as a cofactor. The Biomolecular Effective Fragment Potential (BioEFP) is an *ab initio* force field in which the interaction between the QM and EFP regions is described using both electrostatic and polarization terms. The reduction potential was then calculated using the LRA, achieving a good agreement with the experimental values. When a non-polarizable force field was considered, the authors observed a considerable shift in the predicted values, confirming the importance of including environment polarization in the model.

In a recent work [112], the mechanism of the proton-coupled electron-transfer reaction between nicotinamide adenine dinucleotide (NADH) and a protein-bound flavin (FMN) cofactor was studied by both full quantum-mechanics and QM/MM approaches. In the context of a work with a broader scope, the estimated free energies of the proton and electron transfers were in a reasonable agreement with respect to the experimental findings. However, such an approach was limited by the QM region size as well as by a proper treatment of the structural-dynamical fluctuations of the system.

The redox potential of the NAD/NADH system in aqueous solution was also calculated by means of the PCM approach [113]. In this work, the authors showed that a truly accurate evaluation of the redox potential can be achieved using a high level of theory (i.e., G3B3), thus limiting its application to simple molecules.

#### 3.1.4. Redox potentials of Cytochromes

Under the protein family termed cytochromes, there are different kinds of electron-transfer and redox catalyst proteins, with one or more heme groups as active sites. Multiheme cytochromes are particularly interesting because of their possible use in nanobiotechnology as molecular wires [2,114,115,116,117].

Therefore, to better understand their redox properties at an atomic level of detail, several theoretical approaches have been applied in the past.

Warshel and Churg applied the protein dipoles—Langevin dipoles (PD-LD) [84,118] to cytochrome c (Cyt C) in water in order to model the difference in the reduction potential of the active site (heme) in protein and when it is bound to an octapeptide–methionine complex in water (obtained by the hydrolysis of cytochrome c) [119].

Breuer et al. in [66] investigated the heme-to-heme electron-transfer free-energy landscape of the deca-heme protein MtrF throughout the MD simulation. They considered the reduction of one cofactor at a time, while all the other cofactors were in the oxidized state. The free-energy profile was obtained via thermodynamic integration between the oxidized and the reduced form for all the heme groups. The electrostatic effect of the protein and the solvent was assumed to be the major contribution, justifying the use of classical force fields for the calculation of redox potentials of the cofactors. The authors found a free-energy landscape approximately symmetric with respect to the protein center and suggested a two-dimensional reversible character of the MtrF electron transport under aqueous conditions [66].

In the work of Daidone et al. [55], the reduction potential of the two heme groups of a diheme cytochrome c (DHC; see Figure 5) was estimated by the PMM (see theoretical approaches, Section 2.3.2). The DHC is involved in the electron-transport pathway from a quinol via CytB and the two DHCs to sphaeroides heme protein [120]. The DHC protein has two heme sites, and the four possible redox states were considered. Two possible pathways for the complete reduction were proposed, and reduction potentials for the heme groups were calculated. The good agreement between the theoretical and experimental values suggests the robustness of the approach. Through the application of the PMM/MD method, the available experimental data, and solvent-accessible surface-area calculations, it was possible to define a plausible path leading to the reduction of both heme groups. Two different paths differing in the order of the reduction process of the two heme sites were used to evaluate the reduction potentials. The analysis of the redox-process thermodynamics as obtained by the PMM/MD procedure for the two paths strongly suggests that one of them—characterized by the sequential reduction of the C-terminal heme followed by the reduction of the N-terminal heme—is the favorite route, in agreement with spectro-electrochemical experimental findings. The same computational approach was previously used for the calculation of the reduction potential of the heme group of the yeast iso-1-cytochrome c in solution, highlighting the effect on this property of the reversible opening of two water channels in this enzyme [53].

The successful application of PMM to cytochrome systems in solution prompted the application of the method to the prediction of the reduction potential of these proteins anchored to inorganic surfaces [56,121].

In a recent work by Karnaukh and Bravaya [122], a polarizable embedding QM/MM method was used to predict the reduction potential of the heme group in the cytochrome c peroxidase of *Nitrosomonas europaea*. Using a QM/BioEFP [111] approach, the authors included the polarization effect of the environment in the simulation. The reduction potential was then calculated using the LRA, whose validity was confirmed by the Gaussian distribution of the computed values of Vertical Ionization Energies (VIEs) and Vertical Electron Affinities (VEAs). By comparing the result to the experimental value, a significant overestimation of 0.7 V was observed. This was attributed to an issue in the performance of DFT in the description of an iron center with a partially filled d-shell, which potentially requires to consider the multiconfigurational nature of the electronic states in the heme group.

#### 3.1.5. Redox Potentials of Small Molecules in Enzymes

Although we focused this review on the redox potentials of biomolecules, it is worth noting that theoretical-computational approaches addressed the calculation of reduction/oxidation processes of small molecules as occurring in enzymes. Among them, the quantum-cluster approach [61] has been used, for example, to estimate the water oxidation catalyzed by the photosystem II (PSII) as well as the reduction of nitric oxide in nitric-oxide reductase. The results, briefly described below, are presented to highlight that by means of such an approach, it is in principle possible to use it to calculate the redox potentials of (a part of) biomolecules in complex environments.

Redox potentials of OEC in PSIIThe oxygen-evolving complex (OEC; see Figure 6) in the PSII has been one of the most studied redox active sites to describe the mechanism of water oxidation catalyzed by the PSII. The use of DFT calculations and quantum-chemical-cluster methods [62] have been useful in the structural characterization of the intermediate states of the active site as well as in the estimation of the relative energies of the water oxidation-reaction steps [123,124].Nitric-oxide reductaseNitric-oxide reductase (NOR; see Figure 7) is an enzyme that catalyzes the reduction of nitric oxide to nitrous oxide [125]. To gain insight in the catalytic steps describing such a reaction, the quantum-chemical-cluster approach [62] was applied. Such a method suggested a cis:b3 mechanism with respect to the trans mechanism, which was found to be energetically unfavorable. In addition, the authors described the energetics of several steps involved in this reaction and found the mechanism to be pH-dependent, in agreement with experimental data.

### 3.2. Redox Potential of Nucleic Acids

The calculation of the redox potential of nucleic acids and its components (see Figure 8), e.g., nucleobases, nucleosides, and nucleotides, represents an important challenge in the field. In fact, on the one hand, the human genome is constantly exposed to oxidizing agents that could cause one-electron oxidation of nucleobases in nucleic acids leading to irreversible reactions and mutations [126,127]. On the other hand, the experimental estimates of the redox free energies are often difficult to obtain at physiological conditions.

Therefore, knowledge of the redox potential of nucleic acids and its components may help in the determination of the oxidation-prone sequences and in a deeper understanding of the key players in the genome oxidative-damage process.

As explained in the previous sections, the theoretical-computational determination of the thermodynamic properties of complex molecules in solution is not trivial. In fact, a full quantum-mechanical treatment of the system is usually not possible due to the typical size of the systems, but—at the same time—it is fundamental to properly account for the effect of the environment.

Therefore, different theoretical-computational strategies have been developed, and, in the next subsections, the relevant literature in the context of the nucleic acids’ redox-potential estimate is reported and discussed.

#### 3.2.1. The Redox Potential of Single Nucleotides/Nucleosides

Using an implicit solvent approach, Psciuk et al. calculated the reduction potentials of nucleosides in solution [14]. In that work, thermodynamic cycles are used to obtain the reaction free energies for the redox processes, where the solvation free energies are estimated using the solvation model based on the solute electron density (SMD, “D” in SMD stands for “density” to denote that the full solute electron density is used without defining partial atomic charges) [14,40]. In this kind of approach, a key issue is the definition of the solute cavity, because it represents a low-resolution description (i.e., an approximation) of the true molecular system. In the SMD model, the boundary surface between the solute and the continuous solvent is defined spatially by the surface of the resulting superposition of nuclear-centered spheres with atom-type dependent radii, whose values depend on the atomic numbers of the corresponding atoms. The resulting free energy of solvation calculated using such an approach strongly depends on both the model radii and surface-tension terms, which are obtained from an empirically determined set of parameters [40]. The comparison between that work and the available experimental data [128,129,130,131] shows that in water the reduction potentials of nucleobases are affected by a systematic error, probably due to the implicit solvation treatment of the environment (in acetonitrile, a better agreement with respect to the experimental data was found [132]).

A similar approach was also used by Crespo-Hernandez et al. [133]. In that work, the authors make use of a free-energy cycle to estimate the redox potentials and the ionization energies of a set of organic molecules at the DFT/B3LYP level of theory. The solvent effects were treated by the PCM method [134]. The reversible redox potentials for the selected molecules are known with good accuracy in DMF (*N*,*N*-dimethylformamide) and ACN (acetonitrile) solutions. Because they span a wider range of potentials than those expected for the DNA nucleosides, the observed correlation between the oxidation potentials and the VIEs was used to estimate the reversible oxidation potentials of DNA building blocks (i.e., nucleobases and nucleosides) by using the VIEs calculated at the same level of theory for the set of reference molecules. Using the same procedure, the VIEs and oxidation potentials of Watson–Crick base pairs (A–T and G–C) were computed, obtaining values similar to the experimental data for the redox-potential shift of C–G pairs with respect to the single G base [135].

Thampa and Schlegel [136] found that the improvement in the agreement between the calculated and the measured oxidation potentials of the *N*-methyl-substituted nucleic acid bases was modest when few water molecules were explicitly treated within a SMD implicit solvation scheme.

Wang et al. [137] proposed a QM/MM approach for the calculation of the one-electron oxidation potential of nucleobases and deoxyribonucleosides (dRNs) in water. The authors employed the DFT/B3LYP level of theory for the description of the quantum region (QM) and molecular dynamics for the classical region (MM). They extracted, in the last 1 ns of the MD trajectory, 20 snapshots that have been used in the QM/MM the calculation as the initial geometries. By means of LRA, they calculated the free-energy change in the oxidation process allowing the evaluation of the oxidation potential. The QM region is formed by the nucleobases or dRNs and a shell of water molecules. They observed that the hole was delocalized also over the nearby water molecules. Such a result might explain the deviations between the author’s results and those obtained by Paukku et al., using a similar approach [15]. An additional finding of that work is the solvent’s ability to screen the ribose effect on the ionized nucleobase, resulting in limited differences of the AIE between deoxyribonucleosides and nucleobases in water with respect to the vacuum condition.

A similar solvent effect on the VIE of purine nucleobases, nucleosides, and nucleotides was observed in a previous computational work [138]. In that work, the effect of the solvent (modeled by means of a nonequilibrium polarizable continuum model (NEPCM) [139,140]) on the first VIEs of these species was mainly due to the long-range interaction with the solvent bulk [138]. For Guanine, the solvent effect was to lower the VIE by c.a. 1 eV with respect to the gas phase. In the gas phase, the addition of the ribose to position 9 of the guanine nucleobase had the effect to decrease the VIE of 0.14–0.18 eV, whereas the addition of a ribose 5${}^{\prime }$-monophosphate in the same position decreased the VIE of 1.9–2.1 eV (a range of values was reported because the different tautomers were considered). Introducing the solvent effect within the NEPCM scheme, the differences of the VIE of guanine, its nucleoside, and its nucleotide vanished. This was attributed to the solvent effect, which screens the perturbation due to the ribose and ribose 5′-monophosphate, in line with the experimental findings [141].

In this context, we recently applied the PMM, an approach particularly suitable for the determination of the thermodynamics properties of molecules in complex environments. By means of PMM, we calculated the oxidation free energies of deoxynucleosides (see Figure 9) in aqueous and acetonitrile solutions [16]. From MD simulations of the reduced (B) and oxidized state (B^+^) of nucleosides, we calculated the perturbed electronic energies of the nucleobases (selected as the quantum center of the system) to evaluate their corresponding reduction free energies by Equation (Equation 10).

Our estimates of the reduction free energies, in rather good agreement with the available experimental data, showed the relevant effects of the environment in stabilizing the oxidized state of the nucleobases. The analysis of the perturbation pointed out that (i) a relevant number of water molecules should be considered to assure a proper description of the perturbation, (ii) the estimates of the free energies become accurate after several ns of simulations, and (iii) the presence of additional nucleotides in dinucleotides and of the phosphate groups does not change remarkably the free energy of the process [16]. It is very likely to extend this approach to the estimation of the nucleobase redox potential in more-complex systems, such as single and double strands in solution, and, therefore, studies in this direction are underway in our group.

Although the computed and experimental values of oxidation potentials of nucleobases and their derivatives (nucleosides and nucleotides) in solution span a wide range of values, it is comforting to note that all the works agree in the oxidation order between nucleobases (G < A < T ≈ C).

It is worth mentioning that, beside the calculation of the redox potential, an additional issue concerns the localization of the excess charge in the molecule. A proper treatment of such an aspect is mandatory for a correct selection of the QM region in simple nucleic acids, and it is even more important when more-complex molecules, such as DNA, are investigated (see below).

Therefore, several experimental and theoretical-computational works focused on the localization of the HOMO in nucleotides/nucleic acids.

DFT/B3LYP calculations performed by Yang et al. showed that in vacuum the HOMO for 2${}^{\prime }$-deoxyadenosine 5${}^{\prime }$-monophosphate (dAMP^−^), 2${}^{\prime }$-deoxycytidine 5${}^{\prime }$-monophosphate (dCMP^−^), 2${}^{\prime }$-deoxythymidine 5${}^{\prime }$-monophosphate (dTMP^−^), and 2${}^{\prime }$-deoxyguanosine 5${}^{\prime }$- monop hosphate (dGMP^−^) is located on the negatively charged phosphate group, while for 2${}^{\prime }$-deoxyguanosine 5${}^{\prime }$- monophosphate (dGMP^−^) in the anti-conformation, the HOMO is localized on the guanine nucleobase [142]. For 2${}^{\prime }$-deoxyadenosine 5${}^{\prime }$-monophosphate (dAMP^−^), this finding is supported by Hou et al. [143]. Similar conclusions were obtained by Zakjevskii et al. [144] (except for dTMP^−^, where they predicted the localization of the HOMO on the nucleobase).

In fact, the effect due to the solvation, perturbing the (vacuum) electronic orbitals, localizes the HOMO on the nucleobases, thus making the application of QM/MM methods to nucleobases, nucleosides, and nucleotides in solution feasible. Such a solvent-induced localization effect was observed by using rgw microsolvation approach [145,146] as well as by implicit solvent methods (vide infra). Therefore, these findings justify the intuitive choice to circumscribe the nitrogenous bases as the region of the system to be treated at the quantum-mechanical level. However, the question remains open when dealing with more-complex nucleic acids, because the interaction between nucleobases might affect the electron/charge delocalization.

#### 3.2.2. The Redox Potential of Complex Nucleic Acids
(oligomers/ssDNA/dsDNA)

Although studies on nucleic-acid building blocks contributed to clarify the redox properties of such simpler systems, it is even more interesting to study the electron-transfer processes as occurring in complex nucleic acids such as oligonucleotides, single and double strands of DNA in solution.

However, for this kind of system, the calculation of the redox properties is even more problematic. In fact, on the one hand, the structural–dynamical fluctuations of the systems might affect the redox properties, and, thus, they should be taken into account. On the other hand, it is not completely clear the nature of the electronic hole, which can be localized on a single nucleobase [147,148] or, alternatively, delocalized along the strand [149,150].

This aspect is very important in hybrid methods (QM/MM), where the correct pre-selection of the region of the system treated at the QM level is mandatory. That is, if the oxidized state is characterized by a strong delocalization of the one-electron hole over several adjacent nucleobases, it is crucial to consider a quantum region as large as the hole. The extent of hole localization or delocalization depends on several factors, such as the solvation, the sequence, and the molecular conformation, making the a-priori choice of the quantum region rather difficult [17,151,152,153].

Recently, Kumar et al. calculated the one-electron oxidation properties of ds(5${}^{\prime }$-GGG-3${}^{\prime }$) (see Figure 10) and ds(5${}^{\prime }$-G(8OG)G-3${}^{\prime }$) (8OG = 8-oxoguanine) in B-DNA conformation and the hole distribution in the oxidized products [154]. The calculations made use of integral equation formalism of the polarized continuum model (IEF-PCM) by Tomasi et al. [155] to take into account the effect of the solvent (water), where the electronic properties were calculated at the DFT ωB97XD level of theory. The authors estimated the vertical ionization potentials in (solvent) nonequilibrium conditions (i.e., when only the fast electronic response of the solvent was considered, IP^vert^_NEPCM_), as well as when the solvent was equilibrated (IP^vert^_EQPCM_).

These protocols are termed as nonequilibrated PCM (NEPCM) and equilibrated PCM (EPCM), respectively. The value of the solvent reorganization energy can be obtained from the difference: $\lambda_{1}$ = IP^vert^_NEPCM_− IP^vert^_EQPCM_ and the value of the solute (internal) reorganization energy from: $\lambda_{2}$ = IP^vert^_EQPCM_− IP^adia^. From these quantities, they calculated the one-electron oxidation potentials for ds(5${}^{\prime }$-GGG-3${}^{\prime }$) (1.15 V) and for ds(5${}^{\prime }$-G(8OG)G-3${}^{\prime }$) (0.90 V), in reasonable agreement with the experimental estimate of 1.3 V for a -GGG- stack and 1.18 V for 8-oxoG in monomer form [128].

Considering computational approaches based on a molecular description of the effect of the environment, Diamantis et al. [67] proposed an hybrid QM/MM approach to calculate the redox properties of two large DNA fragments composed of 39 base pairs. They used Car–Parrinello MD for the description of the central triplets, while the rest of the system was simulated by classical MD simulations. From these QM/MM MD trajectories, the authors determined the VIE and the VEA distribution of the QM region in the reduced and oxidized states, respectively. Using a theoretical method that combines Warshel [27,156] and the Marcus theory of the electron-transfer reaction [30,31,32,33], the authors estimated the Helmholtz free energy of the oxidation process and the free energy curves of both the reduced and the oxidized states.

In this context, the PMM has been recently applied to the calculation of the oxidation free energies of deoxynucleosides in aqueous and acetonitrile solutions [16].

Several additional computational works have addressed the estimate of the redox properties of nucleic acids. However, most of them only calculated the vertical or adiabatic energies relative to the electron loss or the electron gain.

In these works, briefly described below, great efforts were devoted to the selection of the quantum region [41,157], investigating the region of the molecule where the electron-transfer process takes place (see previous subsection).

For example, Cauet et al. [158] utilized the nucleobase as a QM region (see Figure 11). This choice was motivated by the electronic-structure calculation on cytidine and the 2${}^{\prime }$-deoxycytidine 5${}^{\prime }$-monophosphate molecule in the gas phase and in DNA performed at the HF/aug-cc-pVDZ level of theory that locates the HOMO on the nucleobase (both in the gas phase and in the DNA environment). For the 2${}^{\prime }$-deoxycytidine 5${}^{\prime }$-monophosphate, the HOMO is localized on the phosphate group in the gas phase and on the nucleobases in DNA. Therefore, the authors calculated the ionization potential of nucleobases in a fully solvated DNA environment (3${}^{\prime }$-TCGCGTTGCGCT-5${}^{\prime }$) using a QM(CCSD(T))/MM method. The QM/MM vertical ionization potentials calculated for bases in the DNA environment showed a large increase of 3.2–3.3 eV compared to the gas phase, due to the electrostatic interactions with the water molecules of the MM region. They performed a series of CCSD(T)/MM calculations for the cytosine embedded in the DNA environment with a different number of water molecules within a certain distance from the DNA helix surface. They modeled the vertical ionization potential as a function of the DNA solvation (from 0 to 40,000 water molecules) and observed an increase in the ionization potential between the range of 0 to 2100 water molecules. Beyond this value, the ionization potential stabilized and converged to the value of 11.56 eV. This trend was also qualitatively observed in our previous work [16], where the same approach was used to clarify the effect of the size of the solvation shell on the VIE of adenosine.

Furthermore, the QM region was expanded to a few nucleobases (single- and double-stranded oligonucleotides d(5${}^{\prime }$-C8G9-3${}^{\prime }$), d(5${}^{\prime }$-C8G9C10-3${}^{\prime }$), and d(5${}^{\prime }$-A7C8G9C10-3${}^{\prime }$)) to include π-π interactions between stacked bases as well as intrastrand H-bonds at a slightly lower level of theory (DFT/B3LYP). The inclusion of few bases in the QM region has the effect of lowering the ionization potential of the sequence region, in line with other computational works [159,160,161]. For these “clusters” of bases, a positive shift in the vertical ionization potential was observed in a fully solvated DNA environment. As reported by the authors: “calculations demonstrate that increased vertical IP values of DNA bases and clusters of DNA bases are due to the presence of an electric potential created by the structure of the MM solvent.” Remarked upon was the importance of the explicit treatment of the solvent, “which is a very significant advantage over the continuum models, to provide a predictive capability for ionization of DNA components”.

In the efforts to describe the excess charge delocalization extent, Kumar et al. [154] calculated the spin-density distribution of one-electron oxidized ds(5${}^{\prime }$-GGG-3${}^{\prime }$) in the equilibrium geometry of the neutral species (vertical ionization) using the ωB97XD-PCM/6-31G** level of theory neglecting the solvent reorganization (NEPCM). The results indicate a delocalization of the total spin-density distribution (35% on 5${}^{\prime }$-G, 62% on the central G, and 3% on the 3${}^{\prime }$-G). On the other hand, the calculation of the spin-density distribution of one-electron oxidized ds(5${}^{\prime }$-GGG-3${}^{\prime }$) in its equilibrium geometry using the ωB97XD-PCM/6-31G** level of theory, with the equilibrated solvent, brought different results: 95% of the total spin density was localized on the 5${}^{\prime }$-G in ds(5${}^{\prime }$-GGG-3${}^{\prime }$) in solution, in agreement with previous works [18,19].

An analogous result was obtained for ds(5${}^{\prime }$-G8OGG-3${}^{\prime }$), where 90% of the total spin density was localized on the 8OG (8-oxoguanine) in its equilibrium geometry and with the equilibrated solvent. These findings indicate that solvation has the effect to localize the hole distribution, as also observed in previous computational works [151,162].

A tight-binding approach was applied for the modeling of the DNA oxidation in water [163]. In that work, the authors proposed a novel set of hole-site energies and intrastrand electronic-coupling parameters for predicting DNA hole-trapping efficiencies and rates of hole transfer in oxidized DNA. The need to consider hole-site energies and electronic-coupling parameters in solution was recognized, and water effects were included via the polarizable continuum model (PCM) [36]. The authors underlined the importance of the parametrization of the tight-binding Hamiltonian with hole-site energies and electron-coupling parameters describing the solvation contribution. In fact, most of the previous efforts to obtain these parameters did not include the solvent effect according to the hypothesis that the relative gas-phase ionization energies of nucleobases should provide a reasonably good estimate of the hole-site energies in solution [163]. However, this argument is not fully supported by a previous work, where it was found that the environment severely affects the ionization energies of nucleobases in solvated DNA [164]. By means of a tight-binding approach, the authors computed the adiabatic and the vertical ionization potentials of guanine (G), adenine (A), cytosine (C), thymine (T), uracil (U), and 6-azauracil (X) in aqueous solution and in a short oligonucleotide: 5${}^{\prime }$-XXYX-3${}^{\prime }$ (where Y = G, A, C, T) at the PCM-B3LYP-D/TZV+P level of theory (the VIEs were computed via the non-equilibrium polarizable continuum model). Hole energies of DNA nucleobases can be estimated by the ionization potentials of 5${}^{\prime }$-XXYX-3${}^{\prime }$ single strands under the condition that the positive charge is localized on Y. They also calculated the adiabatic ionization potentials of 5${}^{\prime }$-XYZX-3${}^{\prime }$ single strands, where Y, Z = G, A, C, T, to predict the intrastrand electron-coupling parameters between the four DNA nucleobases. To test the goodness of the parameters extracted from the adiabatic and VIEs of the nucleobases, they computed, in the selected sequences, the ionization energies of several tetrameric sequences at the DFT level of theory and with the tight-binding Hamiltonian approach. The authors found that the TB Hamiltonian results are in good agreement with the DFT computations, as well as with experimental data [163]. To corroborate the importance of including solvent effects in the parametrization, the authors concluded that “a major difference with previous parameterizations is the incorporation of the electrostatic effects of the surrounding medium, so that hole-site energies reproduce almost quantitatively the results of photoelectron spectroscopy and differential pulse voltammetry”.

## 4. Conclusions

Due to the complexity of biomolecular systems, current theoretical-computational approaches allow the estimation of the redox free energy with different reliability.

Fundamental differences between such approaches mainly arise from the description of the environment as well as of the structural-dynamical fluctuations of the whole system, two factors which can severely affect the free energy of the process.

The main findings of the literature in the field show that the explicit description of the environment, and its interaction with the redox center, greatly improves the reliability of the methods, suggesting that mixed quantum-classical approaches are the most-useful theoretical-computational tools. Furthermore, it is also evident that the delocalization of the excess charge should be carefully considered in the choice of the QM region, and a high level description of the electronic states for the molecules/chemical groups involved in the redox reaction is essential to obtain robust and accurate results. Finally, it is worth remarking that, despite different theoretical treatments used, efficient methods for modeling redox thermodynamics should be highly general (applicable to different systems without ad hoc parametrization) with enhanced usability (allowing their use to a broad scientific community) and significantly robust (allowing for simple reliability tests and easy improvements).

## Figures and Tables

**Figure 1 molecules-27-01077-f001:**
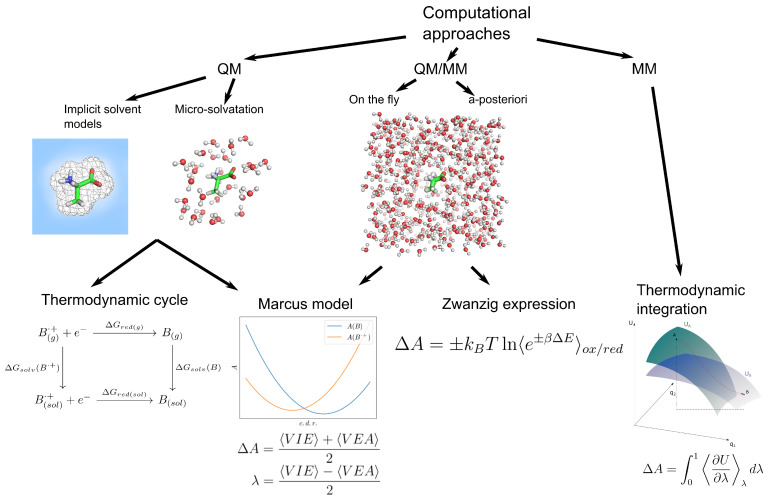
Synthetic scheme of available computational approaches.

**Figure 2 molecules-27-01077-f002:**
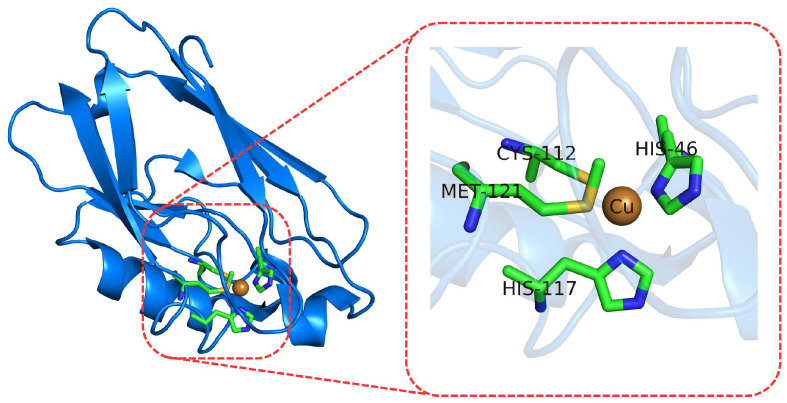
Cartoon representations of the Azurin protein (**left**) from *Pseudomonas aeruginosa* (PDB: 4AZU) and its copper binding site (**right**).

**Figure 3 molecules-27-01077-f003:**
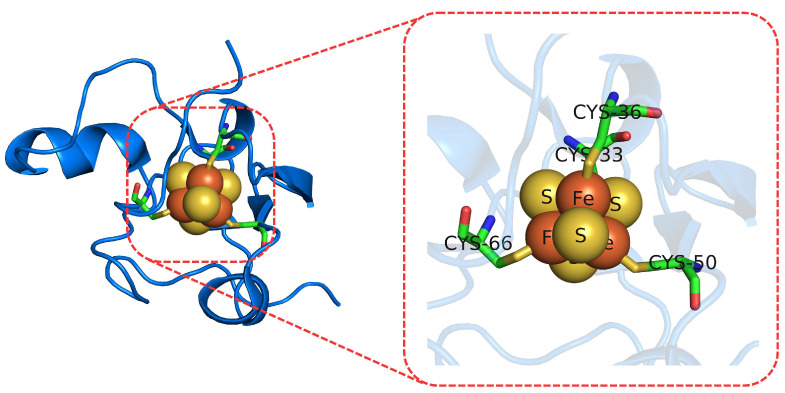
${\mathrm{Fe}}_{4}S_{4}$ cluster (**right**) found in high-potential iron–sulfur proteins I from *E. Halophila* (**left**).

**Figure 4 molecules-27-01077-f004:**
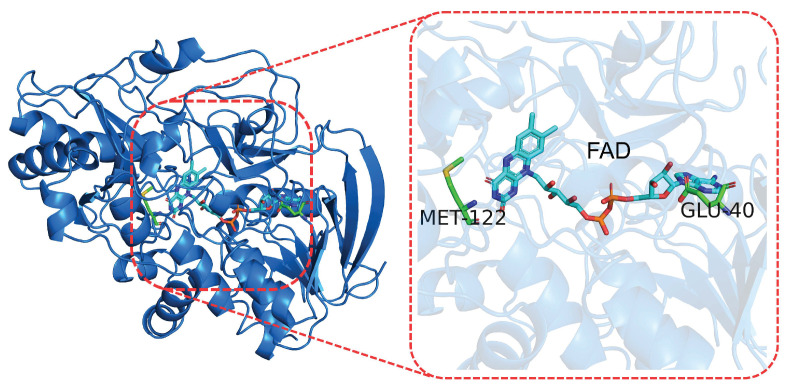
FAD cofactor (**right**) found in cholesterol oxidase from Streptomyces (**left**).

**Figure 5 molecules-27-01077-f005:**
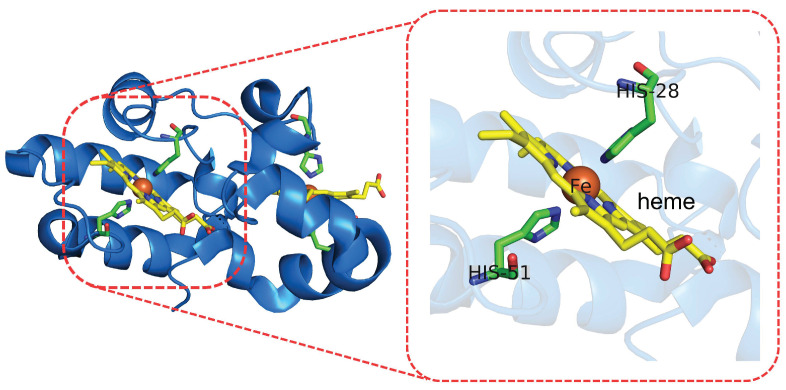
Crystal structure of diheme cytochrome c (DHC) from *Rhodobacter Sphaeroides* (cartoons, **left**) and its active site (**right**).

**Figure 6 molecules-27-01077-f006:**
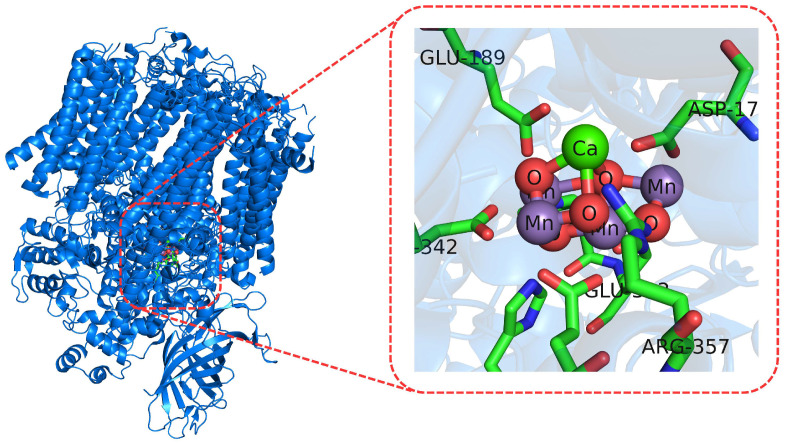
Active site (**right**) of the oxygen-evolving complex in PSII (**left**).

**Figure 7 molecules-27-01077-f007:**
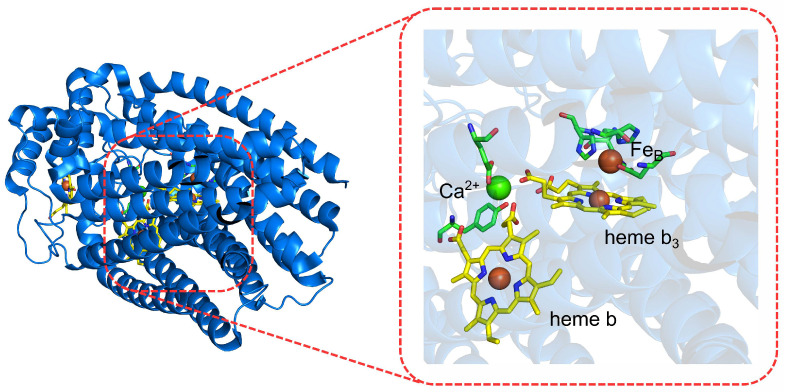
Active site (**right**) of nitric-oxide reductase (**left**) obtained from *Pseudomonas aeruginosa* (PDB: 3WFB).

**Figure 8 molecules-27-01077-f008:**
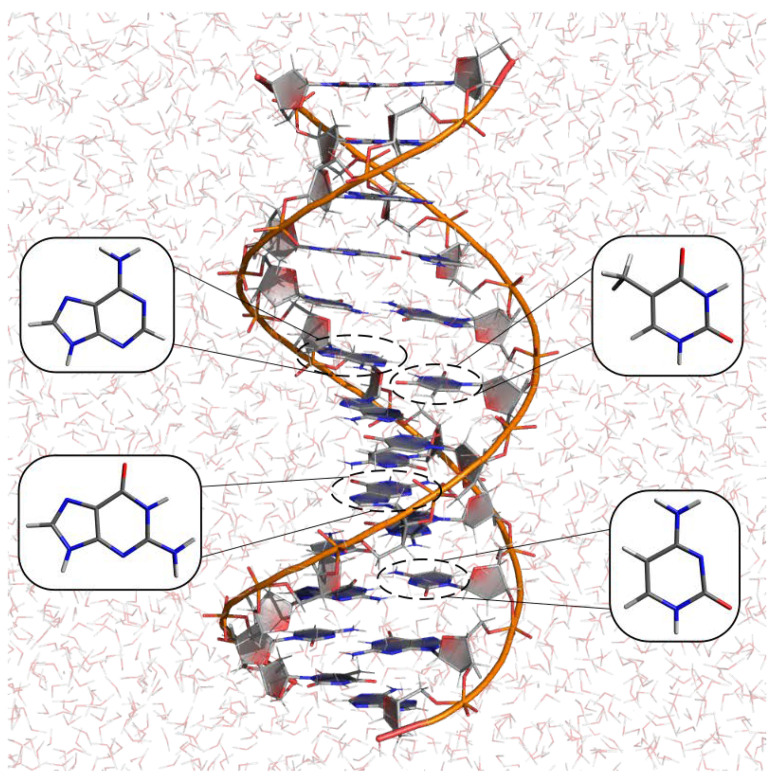
Solvated DNA double strand. The four DNA nitrogenous bases are highlighted.

**Figure 9 molecules-27-01077-f009:**
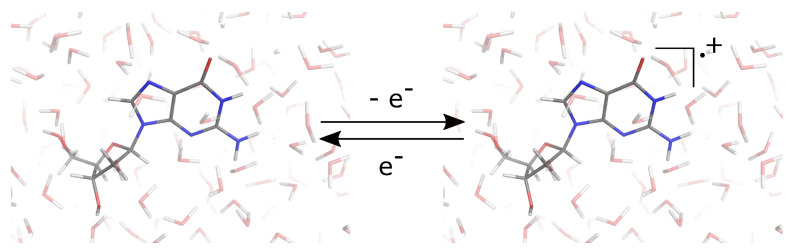
Schematic representation of the redox reaction of the neutral guanosine in aqueous solution.

**Figure 10 molecules-27-01077-f010:**
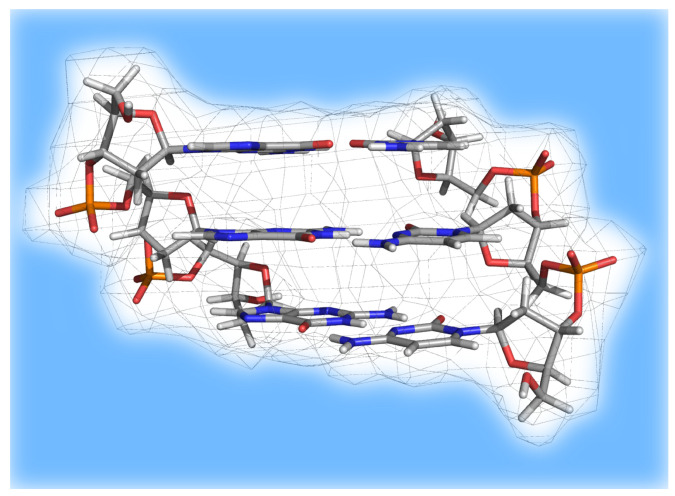
Double-stranded oligonucleotide ds-(5${}^{\prime }$-GGG-3${}^{\prime }$) studied by Kumar et al. [154]. DFT calculations were performed on the atoms represented in stick, whereas the environment (indicated by the the light blue region) was modeled by means of the IEF-PCM implicit solvent model.

**Figure 11 molecules-27-01077-f011:**
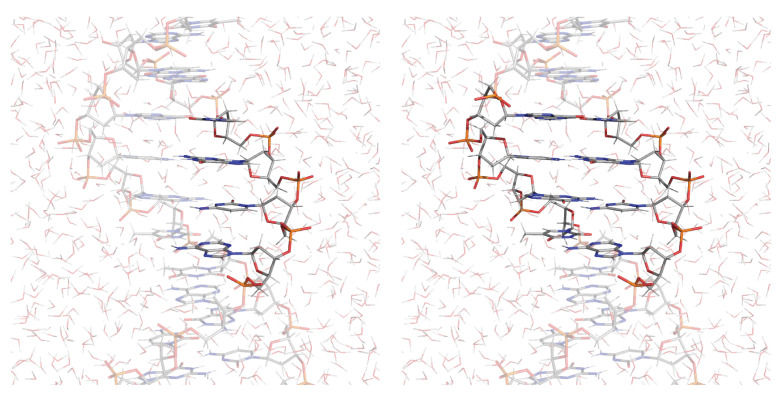
Representation of the solvated ds-(3${}^{\prime }$-TCGCGTTGCGCT-5${}^{\prime }$) DNA double strand studied by Cauet et al. [158]. The highlighted atoms represent the two different QCs chosen in that work.

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
