# Peer review of "Theoretical Modeling of Redox Potentials of Biomolecules"

_molecules, 2022, doi:10.3390/molecules27031077_

Round 1
Reviewer 1 Report
This manuscript reported a summary of computational modeling of the redox potentials of biomolecules, including enzymes. The main focus of the work seems to be mainly one-electron oxidation/reduction, while proton-coupled electron transfer oxidation/reduction was only discussed for the NAD、FAD redox potentials. Overall, the manuscript is of excellent quality, and may be published after addressing the following minor issues:
- The quantum chemical cluster approach developed by Siegbahn, Himo, et. al. has also been used for the modeling of redox potentials, especially for proton-coupled electron transfer oxidation/reductions. Excellent examples can be seen for water oxidation by OEC in PSII, O2 reduction in cytochrome c oxidase, NO reduction in nitric oxide reductase.... (see Chem. Rev. 2014, 114, 3601 ), and also later work on reductive dehalogenase (ACS Catal. 2015, 5, 7350). This approach and the related results could be discussed as well.
- Some figures could be added to explain and compare the theoretical approaches, and also the examples discussed.
- The error bar of different approaches could be discussed.
Reviewer 2 Report
The article provides a comprehensive review on the computation of the redox potential of biomolecules (proteins and nucleic acids). Not only historical works, but also the current challenge in calculating the free-energy difference between the reduced and oxidized states is clearly illustrated, namely, the treatment of the environment (QM/MM, the size of QM region, the level of QM method), and the structural sampling. The review broadly covers the advances of the field in the recent 20 years with good balance. I therefore recommend the publication of the manuscript to Molecules.
I have only a few comments.
- The authors have developed an original QM/MM-like method, perturbed matrix method (PMM), to represent the system, and applied the method to calculate the redox potential. I think the PMM method is a unique scheme that deserves to be described with more emphasis and details. I highly appreciate the authors’ writing that introduce not only their own works, but also many from other groups. Nonetheless, highlighting the PMM method will make the review more unique and attractive.
- Specifically, the PMM is described in page 6, lines 245 – 258. Although the references to the original articles are given there, it would be better to expand this paragraph and describe in more detail the basic equations and the actual procedures of calculation.
- In Section 3, again I recommend the authors to give the result of the PMM method with more details in page 8, lines 359 – 370; page 11 lines 487 – 500; page 13 lines 594 – 605; page 14, lines 678 - 680. Besides technical aspects, more discussions on scientific insights that the method has brought about would be appealing. In particular, the complex nucleic acids would be intriguing for me. The use of figures to explain the systems and the main finding is helpful to understand.
The followings are minor comments.
- In page 3, what is the typical length of MD simulation required for a sufficient sampling? Pico-sec, nano-sec, or micro-sec? Of course, it should depend on the system, but it is helpful to know what’s the general consensus in the field.
- Section 2.3.1 is a bit scattered and needs more structured paragraphs. Please polish up the writing.
- In page 13 line 626, the effect of solvation on the electron localization is very interesting. Is such an effect described by the implicit model?
- Typos:
Page 10, line 480: termodinamic -> thermodynamics
Page 11, line 489: What is DCH”xmgra”?
Page 11, line 495: method was -> method, it was
Page 11, line 509: VIE (vertical ionization energy) and VEA (vertical electron affinity) They appear for the first time here.
Page 13, line 605: areunderway -> are underway
Page 13, line 630: quantomechanical -> quantum mechanical
Page 13, line
- The authors may find following references also interesting the present context.
Consistent scheme for computing standard hydrogen electrode and redox potentials,
Matsui et al., J. Comp. Chem. 34, 21 (2013).
https://onlinelibrary.wiley.com/doi/10.1002/jcc.23100
Accurate Standard Hydrogen Electrode Potential and Applications to the Redox Potentials of Vitamin C and NAD/NADH,
Matsui et al., J. Phys. Chem. A 119, 369 (2015).
https://pubs.acs.org/doi/10.1021/jp508308y
Reviewer 3 Report
The authors have attempted to put together the work done in the field of theoretical methods used in modelling redox potential in biomolecules. It is a very challenging, interesting and a dynamic field which is constantly evolving to improve the theoretical prediction of redox potential in biomolecules. The authors have touched all the essential methods that are used in these calculations and it is an interesting and informative read. I have the following recommendation to the authors; my attempt was to try to add aspects to the review that will help people who are not as experienced as the authors in this field to understand the complexity of these calculations.
The authors talk about several protein families like P450, cytochrome c, hydrogenase, copper proteins etc that play essential role in maintaining the redox potential essential for electron transfer in proteins. They also mention about cofactors like NAD/FAD/FMN as well as the nucleic acid systems. Readers may not be familiar with all the systems the authors are talking about. To me, as a reader, I would like to visualize the systems and have an idea of redox site to relate to the complexity of these calculations. Since it is a review and intended to reach out to wider audience, I think it might be worthwhile to provide some figures of the systems mentioned and highlight the redox center, either when defining them in the introduction or where the case studies are discussed, wherever the authors think is most suited.
Since it is a review I would like the authors to expand the theoretical section (section 2) to include a bit of description in the context of how these theoretical approaches are used in methods used for calculating redox potential even if there is a bit of overlap with this section and the case studies. For example, in the section 2.3, I think the authors gave a very general view of the methods but failed to emphasise how these methods are incorporated in context of redox calculation. Similarly, it applies to the implicit solvent approaches too. I think this would help when going back to the case studies where several methods which apply these theoretical approaches are described case by case.
Minor corrections:
There are several places where the abbreviation is used before the terms are described (e.g. VIE, VEA), some of the terms are defined several times and re-abbreviated (e.g. VIE, PMM, SMD). Please make sure that the terms are abbreviated first time they are described and the abbreviations are used uniformly throughout the review.
Change “adiacent” to adjacent in lines 55, 61 in page 2 and line 647 of page 14
Kindly check the grammar on line 60 - suggestion - "can affect" to "affects"
Line 65 – add: …the role “of” redox potential “in” bio-macromolecules – please recheck.
Line 80 – “dishomogenous” to “inhomogeneous” – kindly check for appropriate word
Line 192 – I am not sure "remanding" is the appropriate word here. Maybe directing or pointing?
Line 198 – shouldn’t it be “empirical” instead of “semi-empirical” if the authors are referring to the standard force fields?
Line 221-222 – As far as my understanding is - In polarized embedding, in addition to polarization of the QM (inner) region by the outer MM charges, the polarization of the outer MM region by the presence of the inner QM region is also considered. This is not clearly coming out by the way the authors have described here.
Line 438 – “The computational approach used..” could the authors mention here in short what the method is and kindly check the reference provided in that paragraph and to which of the method it describes?
Line 457 – “quantum-mechanic” to “quantum-mechanics”
Line 468 – correct the spelling to “catalyst”
Line 480 – correct the spelling to “thermodynamic”
Line 500 – “channels on this property” – may be change it to “..channels in this enzyme”
Line 605 – space between are and underway
Line 732-733 - “..effect of localize” to “effect to localize”
Kindly do a spell check of the document so that the typographical errors are eliminated.
Round 2
Reviewer 3 Report
I thank the authors for considering all the points raised and addressing them.
Here are my few comments:
I thank the authors for adding the figures, however, it would be nice if they could refer the figures in the text where possible to add them in context.
Lines 167 - 171: “The redox potential of molecules by means of implicit solvent…“ – two methods of evaluation are stated; the thermodynamic cycle and Marcus model and Figure 1 is referred. In Figure 1, however, as per my understanding, it only refers to the thermodynamic cycle - kindly make the figure consistent with this statement
I am a bit lost reading section between lines 571 to 601 - The review is on modelling redox potential and I really don’t see that aspect in the cases described in this section. I might have missed the point the authors are trying to relay. I would request the author to give a thought about this section so that it justifies the main subject matter of the review.
Some comments on abbreviations:
Line 255: Please add the abbreviation PMM after “ approach, the Perturbed Matric Method,..” because in line 272 “..PPM scheme,..” where I guess PPM refers to Perturbed Matric Method. In the same context, in my opinion, the abbreviation PMM could be used instead of redefining it again at line 407, 538, 683 and 775.
Line 268: Please define QC before using the abbreviation QC
The abbreviation LRA for Liner response approximation is mostly throughout the text once it had been abbreviated on line 124, so in my opinion LRA could be used in lines 391 and 661 instead of its full form.
Thermodynamic integration is used without abbreviation except for line 484 (TI-like) - so maybe just use the full form throughout the text.
Line 530: Will it not be thermodynamic?
VIE are already abbreviated in line 565 so in my opinion using just the abbreviation should be sufficient in lines 646 and 769.
Vertical electron attachment is used once in line 770 so I see no need to abbreviate it, but I leave it to the authors to decide.
